# Switched-capacitor-convertors based on fractal design for output power management of triboelectric nanogenerator

Wenlin Liu [1,4], Zhao Wang[1,4], Gao Wang[1,4], Qixuan Zeng[1], Wencong He[1], Liyu Liu[1], Xue Wang[1], Yi Xi[1], Hengyu Guo[1,2,3 ✉], Chenguo Hu [1 ✉] & Zhong Lin Wang [2,3 ✉]

Owing to the advantages of integration and being magnet-free and light-weight, the switched-capacitor-convertor plays an increasing role compared to traditional transformer in some specific power supply systems. However, the high output impedance and switching loss largely reduces its power efficiency, due to imperfect topology and transistors. Herein, we propose a fractal-design based switched-capacitor-convertors with characteristics including high conversion efficiency, minimum output impedance, and electrostatic voltage applicability. As a double-function output power management system for triboelectric nanogenerators, it delivers over 67 times charge boosting and 954 W m$^{-2}$ power density in pulse mode, and achieves over 94% total energy transfer efficiency in constant mode. The establishment of the fractal-design switched-capacitor-convertors provides significant guidance for the development of power management toward multi-functional output for numerous applications. The successful demonstration in triboelectric nanogenerators also declares its great potential in electric vehicles, DC micro-grids etc.

[1] Department of Applied Physics, State Key Laboratory of Power Transmission Equipment & System Security and New Technology, Chongqing University, 400044 Chongqing, P. R. China. [2] Beijing Institute of Nanoenergy and Nanosystems, Chinese Academy of Sciences, 100083 Beijing, P. R. China. [3] School of Materials Science and Engineering, Georgia Institute of Technology, Atlanta, GA 30332, USA. [4] These authors contributed equally: Wenlin Liu, Zhao Wang, Gao Wang. ✉email: cquphysicsghy@126.com; hucg@cqu.edu.cn; zhong.wang@mse.gatech.edu

As a highly efficient device in low-frequency energy harvesting[1–3], triboelectric nanogenerator (TENG) has been demonstrated with great potential for applications of self-powered sensor system in biomedical science[4,5], chemistry[6,7], smart devices[8–10], and mechanical energy harvesting from human movement[11,12], ocean[13,14], wind[15,16], etc[17–24]. Based on the coupling of triboelectrification and electrostatic induction[25], TENG can easily generate several kV high voltage[26], however, the low current (~μA), low charge density (~100 μC m$^{-2}$) and high impendence (1–100 MΩ)[27,28] causes extremely low energy utilization efficiency[29]. Thus, high efficient power management system is urgently needed for maximally utilizing energy generated from TENG. In previous researches[29–44], a traditional transformer is applied to TENG directly to realize low voltage output, but high operating frequency is necessary[30–33]. Later, strategy of combining an electric switch and an inductance transducer is suggested, and over 80% efficiency is reached, but additional energy is consumed by the electric switch[29,34,35]. Recently, mechanical switch strategy for an auto-switch between serial and parallel connections of capacitors is designed to enhance output charges, but it is difficult to achieve a high transformer efficiency due to its complex mechanical structure[38,39]. Moreover, the current researches of power management only focus on enhancing energy transfer efficiency in a single-function unit, although multi-function units are increasingly important to satisfy various practical demands nowadays. Therefore, a power management system with higher transfer efficiency and multi-function output mode is urgently needed and has great significance for practical applications of TENG.

With the rapid development of energy harvesting technology, more requirements are proposed for energy conversion devices. The traditional transformer, which is widely used as a core component in mobile chargers and power distribution systems is hard to meet present demands owing to insurmountable electromagnetic noise, large volume and empty load power loss[45,46]. Based on MOSFETs and capacitors, a switched-capacitor convertor (SCC) can effectively realize step-down or step-up conversion by switching serial-to/from-parallel connection and phase modulation. With tendency of high frequency and integration[47–49], SCC has displayed great potential for wireless sensors networks[47,48], DC micro-grids[45], electric vehicle[50,51], solar photovoltaic systems[52,53] etc., due to its merits of magnets-free, high conversion efficiency and light weight. Nowadays, the research emphasis of SCC focuses on the topology configuration structure and modulation method to enhance conversion efficiency and expand functions. The basic switched-capacitor topology configuration can be divided into series-parallel, dickson, fibonacci, and ladder[54,55], but the high output impedance, low transformer ratio and high output ripple limit its practical applications. Thus, more topological configurations evolved from basic one are suggested to adopt to various application scenarios. Recently, successive approximation topology configuration and recursive topology configuration are proposed to achieve 117 converter ratio and wide voltage input/output range (0.1–2.2 V)[56,57]. However, the high output impedance loss and large switching loss of SCC is still the main factor that decreases the power efficiency, and thus the SCC is hard to be applied to high electrostatic voltage energy conversion.

Here, considering the high electrostatic voltage of TENG, inspired by fractal theory and serial-parallel SCC, we put forward a fractal design based switched-capacitor-convertors (FSCC) with over 94% conversion efficiency and step-down function. Compared with a basic SCC, the FSCC has advantages of minimum output impedance, high electrostatic voltage applicability and high step-down ratio. A highly effective power management system is also built for TENG coupled with FSCC. By integrating FSCC power management system on a printing circuit board (PCB), over 67 times charge boosting, 14.3 A m$^{-2}$ current density and 954 W m$^{-2}$ power density are reached by a common TENG under a pulse output, and electric devices like buzzer are driven by TENG with the power management system. Under constant output mode, over 94% total energy transfer efficiency is realized with output power of 37.09 mW m$^{-2}$, and mobile electric devices like digital vernier caliper and temperature hygrometer can be driven continuously by the TENG with FSCC power management system as well. The successful demonstration in TENG presents the great prospect of FSCC in electric vehicle, DC micro-grid area etc. The establishment of double-function output power management system for TENG also indicates great applications in internet of things and has significant guidance on the next stage development of power management system toward multi-function output mode.

## Results

**The fundamental concept of FSCC.** For a better understanding of a FSCC, comparison of working principles between a conventional electromagnetic transformer and a FSCC is given. Figure 1a shows the conventional electromagnetic transformer, basic 2 = 2 unit SCC with series-parallel topology configuration and 8 = 2 × 2 × 2 unit FSCC based on fractal design. The conventional electromagnetic transformer consists of primary coil, secondary coil and magnetic core, which can convert AC high voltage and low current into AC low voltage and high current (Step-down) based on current magnetic effect and electromagnetic induction. The step-up effect can be realized as well if we exchange the input and output direction. In the ideal case, the input power equals output power during the process without power loss. Similarly, the basic SCC (consists of capacitors and switches and the switch is commonly achieved by MOSFET) can convert DC high voltage and low charges into DC low voltage, and high charges by charging capacitors in serial connection and discharging from the charged capacitors in parallel connection (step-down) based on the on-off effect of MOSFET. When the SCC charging in parallel and discharging in serial, the step-up effect can be achieved too. The input energy equals output energy in ideal situation on the right of Fig. 1a. The basic series-parallel SCCs can form a FSCC by fractal design, which can achieve better step-down/up by more effectively switching the connection mode of capacitors.

Considering the low charges (~200 nC) in TENG, large switching loss and zero gate voltage drain current (>15 μA) of MOSFET and super-low leakage current (<1 nA) of rectifier diode, the FSCC composed of rectifier diodes and capacitors is designed to convert electrostatic voltage of TENG. Supplementary Fig. 1 shows photographs of 5 W commercial step-down transformer and designed step-down FSCC, and Supplementary Table 1 displays the corresponding parameters. Compared with a commercial transformer, basic SCC has the merits of magnet-free, integration, and light weight. Furthermore, high step-down ratio, minimum output impendence, higher conversion efficiency (>90%) and electrostatic voltage applicability are achieved by FSCC in this work. Inevitably, there is an energy loss mainly caused by output impedance of SCC in an actual conversion process, which is the total output voltage drop caused by diodes during the discharging process. The loss will be large if there are many stages of diodes as switches. The basic SCC has large output voltage drop loss and leads to low energy conversion efficiency. Consequently, a fractal design is used to effectively decrease the total output voltage drop loss for high efficiency.

Fractal means that a rough or piecemeal geometric shape can be divided into several parts, and each part (at least

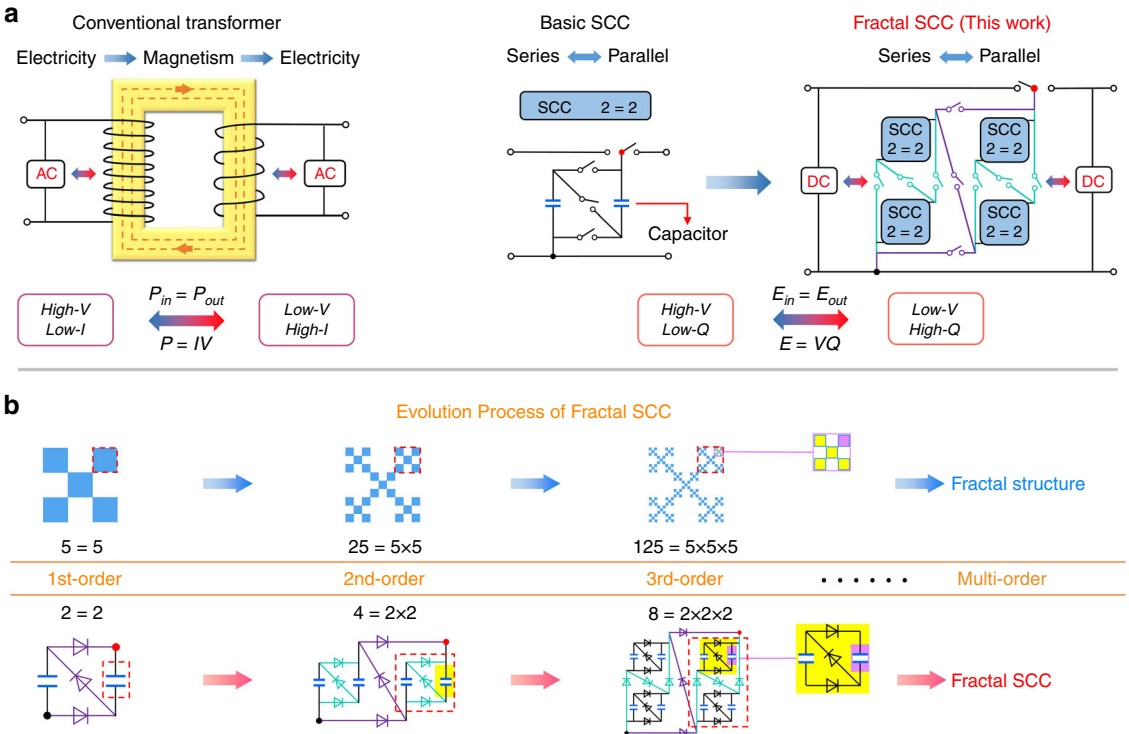

**Fig. 1 Comparison of the two types of transformers. a** The operating mechanism of a conventional electromagnetic transformer, a basic switched-capacitor convertor (SCC) and a fractal design based switched-capacitor-convertor (FSCC). **b** The concept of fractal structure evolution in order increasing and below it is the similar design based fractal structure evolution for FSCC in order increasing, which can be achieved by replacing capacitors in SCC repeatedly with 2 = 2 SCC unit.

approximately) is the shape of reduced whole, that is, it has the self-similar property. Figure 1b shows the concept of fractal structure evolution in order of increasing and below it is the similar design based fractal structure evolution for FSCC in order of increasing, which can be achieved through multiple self-replacement with basic SCC units. In fractal structure evolution, a basic square pattern (consists of five small squares) replaces its part of small square repeatedly and a complex square pattern is formed. The part of the complex pattern is similar to complex pattern itself. For convenience, in the evolution process from basic SCC to FSCC, the basic SCC is defined as 1st order FSCC. Similarly, the 1st order 2-unit FSCC can develop the 2nd order 4-unit FSCC by replacing the capacitors of the 1st order 2-unit FSCC. Multi-order FSCC with different units can achieve in the same way, and the formed FSCC has the self-similar property at last (lower). Supplementary Fig. 2 shows 1st, 2nd, and 3rd order FSCC with 8 units.

In the next part, we introduce the name rules of FSCC to understand it better. First, for the 1st order N unit FSCC, which consists of a number of N basic units. When $N = 1$, there is only one capacitor in the FSCC without step-down function. Thus, there is practical significance only when $N \geq 2$. One basic unit of FSCC consists of one capacitor and capacitors are connected by diodes. The detailed structure of FSCC is described with an expanded form:

$$N_L = m_L \times m_{L-1} \cdots \cdots m_2 \times m_1 \qquad (N \geq 2) \qquad (1)$$

$L$, $N_L$, and $m_i$ belong to integer. $N_L$ is the number of basic unit (including $N_L$ capacitors) and $L$ is the order of FSCC. $m_L$ means the $L$th order $N_L$ unit FSCC that consists number of $m_L$ $(L-1)$th order $N_{L-1} = m_{L-1} \times m_{L-2} \cdots \times m_1$ FSCCs. Each $(L-1)$th order FSCC consists number of $m_{L-1}$ $(L-2)$th order $N_{L-2} = m_{L-2} \times m_{L-3} \cdots \times m_1$ FSCCs, and at last, each 2nd order

$N_2 = m_2 \times m_1$ FSCC consists number of $m_2$ 1st order $N_1 = m_1$ FSCCs. With the expanded form, we can know the number of basic unit, order of FSCC and number of $i$th order FSCCs clearly.

**Operating mechanism of FSCC.** Simply, we chose a $4 = 2 \times 2$ (2nd order 4-unit) FSCC that consists of nine diodes and four identical charge storage capacitors to illustrate its step-down mechanism. Figure 2a shows the initial state of FSCC system. The system consists of a voltage source, switch S1, S2 and a $4 = 2 \times 2$ FSCC. There is no charge in capacitor and the corresponding capacitance is C. When S1 turns on, voltage V from the voltage source is applied to FSCC, correspondingly, input charge quantity of Q is charged into FSCC with charge storage capacitors in series (Red route in Fig. 2b). Charge quantity of Q and voltage of $V/4$ is obtained in every charge storage capacitor. When switch S1 turns off and S2 turns on, the charge storage capacitors are in discharge state and connected in parallel automatically (Red route in Fig. 2c) due to the unidirectional conductivity of diode. And the output charge quantify of $4Q - 4Q'$ and voltage of $V/4 - 2V_d$ are released, where $Q'$ and $2V_d$ are charge and total output voltage drop ($V_{d,t}$), which remains in every charge storage capacitor due to the turn-on voltage drop of diodes and $V_d$ is the voltage drop of one diode. Because $Q'$ remains in capacitor, there is only $Q - Q'$ input charge quantity charging into FSCC when switch S1 turns on again, and $4Q - 4Q'$ is discharged when switch S2 turns on again, respectively. Besides, the operating process of $4 = 4$ (1st order 4-unit) FSCC is shown in Supplementary Fig. 3a–c, which indicates that 1st order and multi-order FSCC both have step-down ability. Compared with basic SCC (1st order FSCC), the capacitors in multi-order FSCC can have minimum total output voltage drop $V_{d,t}$ (charge flowing through the least diodes) due to the 2D diodes network based on fractal design during the discharging process.

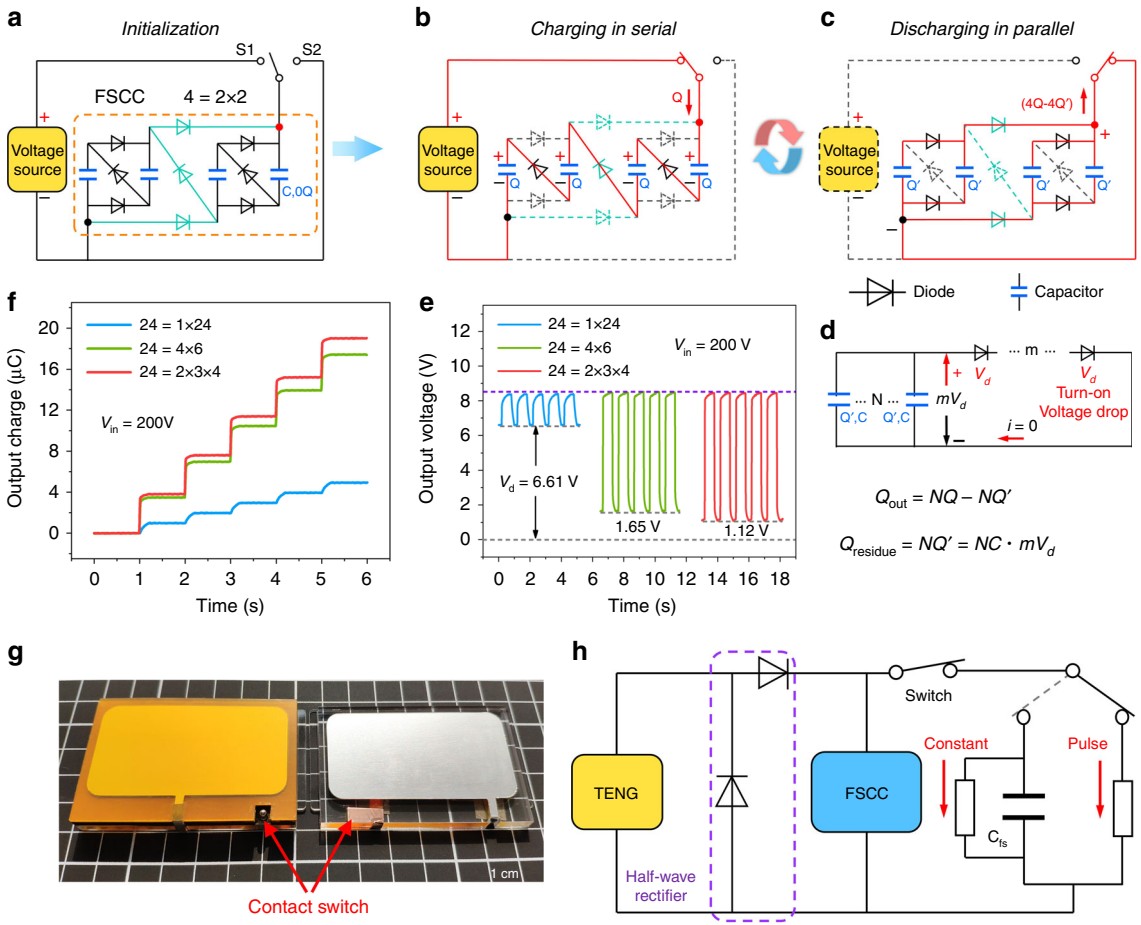

**Fig. 2 The principle of FSCC power management system. a** Initialization state of 4 = 2 × 2 FSCC system. **b**, **c** The step-down process during a charging/discharging cycle. The FSCC can achieve multifold charge output through the auto-switch of capacitors from serial to parallel connection. **d** The equivalent diagram of discharging process of FSCC in **c**. The turn-on voltage drop of diode has a significant influence on charge output. **e** The output voltage and **f** the output charge of 24-unit FSCC with different orders. 1st order FSCC has the largest turn-on voltage drop and the output charge increases rapidly with multi-order FSCC. **g** Photograph of triboelectric nanogenerator (TENG) with a contact switch in this work. **h** Schematic diagram of the power management system, which consists of a TENG, half-rectifier circuit, contact switch and FSCC, and there are two output modes in pulse and constant after power management.

As shown in Fig. 2d, the discharging process of N unit FSCC is equivalent to N charge storage capacitors in parallel as a positive charge source, and m diodes are in series as a negative voltage source due to its voltage drop, which leads to $mV_d$ total output voltage drop and $NC \cdot mV_d$ ($Q = CV$) charges remain in charge storage capacitors. Therefore, there is $NQ - NC \cdot mV_d$ charge releasing from FSCC. The voltage drop-current curve of one 1N4007 diode (Supplementary Fig. 3d) shows that voltage drop (1 V at 1 A for one diode) has a large influence on output (~10 V) of FSCC. The total output voltage drop $V_{d,t}$ under 1st order FSCC is discussed in detail next. Supplementary Fig. 3e displays the theoretical $V_{d,t}$ of the 1st order FSCC, which has a complete linear relationship with the unit number, and the $V_{d,t}$ is further verified by replacing charge storage capacitor with voltage source to measure the $V_{d,t}$ of different units of the 1st order FSCC (Supplementary Fig. 4a, b).

From the analysis above, the effective output charge and output voltage of FSCC are described by equations as follow:

$$V_{out} = \frac{V_{in}}{N} - V_d - V_{d,t} \quad (2)$$

$$Q_{out} = V_{out} \cdot C_s N \quad (3)$$

where $V_{in}$ is the input voltage, $V_{out}$ is the output voltage and $Q_{out}$

is output charge of FSCC, $C_s$ is the charge storage capacitor in FSCC. Detailed theoretical analysis of FSCC is given in Supplementary Note 1. Obviously, the actual convertor ratio $V_{in}/V_{out}$ is smaller than the number of unit due to voltage drop loss.

To obtain exact output characteristics, a voltage source is used to quantify FSCC output performance as shown in Supplementary Fig. 4c. Supplementary Fig. 4d shows circuit and corresponding photograph of 24 = 24 FSCC (total output voltage drop is $23V_d$). The output voltage under 200 V input voltage and output charge under 200 nC input charge of 1st order FSCC with different units, are shown in Supplementary Fig. 4e–g, demonstrating powerfully the step-down ability of FSCC and the great negative influence of $V_{d,t}$ on the output for 1st order FSCC. In order to achieve efficient energy transmission, the fractal property is considered to redesign the switched-capacitor-convertor to effectively reduce the influence of voltage drop. With 2nd and 3rd order 24-unit FSCC, total output voltage drop $V_{d,t}$ can be effectively reduced to $8V_d$ and $6V_d$ in theory. Correspondingly, the experimental output is enhanced by three times, and the $V_{d,t}$ decreases from 6.61 to 1.12 V (Fig. 2e, f and Supplementary Fig. 5). In summary, fractal design can effectively enhance conversion efficiency of SCC.

**Minimum total output voltage drop for FSCC.** For $L$ order $N$ unit FSCC, the relationships of $L$, $N$ and total output voltage drop $V_{d,t}$ (output impedance of FSCC) are described by the following equations:

$$N = \prod_{i=L}^{1} m_i \qquad (4)$$

$$V_{d,t} = \left( \sum_{i=L}^{1} m_i - L \right) \cdot V_d \qquad (5)$$

To obtain the minimum total voltage drop $V_{d,t,\min}$ of FSCC, optimum parameter of $m_i$ is discussed, which is shown in Supplementary Note 2 and Tables 2 and 3. Eventually, and a judgment criterion is obtained as follow,

$$\text{SUM}_{\min} = \left( \sum_{i=L}^{1} \left( m_i - \sqrt[L]{N} \right) \right)_{\min} \rightarrow V_{d,t,\min} \qquad (\text{SUM} \geq 0) \qquad (6)$$

For a $L$ order $N$ unit FSCC, we can see that $m_i$ is closer to $\sqrt[L]{N}$, and $V_{d,t}$ is smaller based on result above. Obviously, when $m_i = \sqrt[L]{N}$, the SUM is equal to 0, there must be smallest $V_{d,t}$ at this situation. And $V_{d,t}$ of $N = 2^L$ and $N = N$ FSCC is listed in Supplementary Table 4, showing that multi-order FSCC can decrease $V_{d,t}$ effectively, and 10th order FSCC can decrease $V_{d,t}$ by over 100 times compared with 1st order.

**FSCC power management system for TENG.** Herein, a highly efficient power management system of FSCC is built for TENG to utilize energy more effectively. Figure 2g shows the photograph of TENG with a contact switch. The 3D structural scheme is depicted in Supplementary Fig. 6a, in which the foam is used to ensure good contact of the switch, and the mechanical switch is used to control the charging/discharging process of FSCC. Figure 2h shows the FSCC power management system of TENG. For the energy E, $E = VQ$, where $V$ is the voltage and $Q$ is charge quantity. Evidently, the energy output can be elevated by boosting the output voltage of TENG. Recently, a method of triboelectric charge supplementary approach that can enhance voltage of TENG is reporte in ref. [20], inspired by that, we use a half-wave rectifier to obtain the maximum energy output.

In our daily life, there are pulse power devices like LED, buzzer. Besides, there are constant current devices like temperature hygrometer and counter. According to these demands, FSCC power management with constant/pulse output for TENG is proposed. A filter storage capacitor $C_{fs}$ is used to obtain the constant output. The working principle of FSCC power management system for TENG is shown in Supplementary Fig. 6b–d and Supplementary Note 3.

**Pulse output of FSCC power management system.** With the development of a super-high power device, such as electromagnetic launcher, pulsed laser, buzzer etc., high pulse power supply is needed to drive them. Therefore, it is significant to study the pulse output after power management, and Supplementary Fig. 7a shows the diagram of FSCC power management system with the pulse output. To characterize the output performance of FSCC power management qualitatively, the preliminary charge quantity of TENG with area of 20 cm$^{-2}$ is controlled at 200 nC, and the corresponding open-circuit voltage reaches 2.3 kV with half-wave rectifier circuit (Supplementary Fig. 7b, c).

As for the short-circuit output charge, the charge storage capacitor (Supplementary Fig. 8) is discussed to obtain the maximum output charge. A capacitor with 22 nF is used in FSCC

to achieve a large output charge and suitable output voltage. Figure 3a indicates that the output charge increases rapidly with the increase in orders and units of FSCC, and the corresponding parameters of FSCCs are shown in Supplementary Table 5. The output charge increases by 50 times from 200 nC to 10.05 μC with $96 = 2 \times 2 \times 2 \times 2 \times 2 \times 3$ FSCC integrated on breadboard (Size of $15 \times 188 \times 280$ mm$^3$), the circuit and photograph of which are shown in Supplementary Figs. 9 and 10a–c. In order to facilitate practical applications, $96 = 2 \times 2 \times 2 \times 2 \times 2 \times 3$ FSCC is integrated on both sides of a $4 \times 60 \times 85$ mm$^3$ PCB as shown in Supplementary Fig. 10d. The output charge increases by 62.5 times and reaches 12.51 μC from 200 nC per cycle (Fig. 3b). It is larger than that on the breadboard owing to the metal foils (Supplementary Fig. 10e) in the breadboard which form many micro capacitors to decrease the output charge. When the charge quantity of TENG reaches 259 nC, the charge output increases by 67.8 times and reaches to 17.55 μC with the FSCC on PCB per cycle (Supplementary Movie 1). The input charge, output voltage and input voltage are shown in Supplementary Fig. 11a–c. The results above display a bigger charge quantity of TENG can reduce the impact of total output voltage drop to realize bigger charge boosting with the FSCC on PCB. In addition, when charging a 100 μF capacitor with FSCC on PCB, the voltage of capacitor can reach 8.76 V from 0 V within 85 s. Meanwhile, the output voltage of FSCC increases from 9.9 V to 14.3 V as shown in Fig. 3c, which indicates a unique voltage cumulative effect and better output stability compared with works based on inductance transducer. And the voltage cumulative effect is caused by the accumulated residual charge in charge storage capacitors ($V = Q/C$) during each discharge cycle, and the residual charge is owning to that the external load is too large to release all the charge. Moreover, even though the operating frequency is as low as 0.1 Hz, over 14.8 μC output charges still retains (Supplementary Fig. 11d), which shows an excellent low-frequency output performance.

As for the pulse output power of FSCC power management system, a maximum pulse output power is reached by employing a $6 = 2 \times 3$ FSCC integrated on PCB (Fig. 3d and Supplementary Fig. 12a). Besides, when the charge quantity of TENG is 291 nC at frequency of 1 Hz and the energy storage capacitor is 1 nF, the short-circuit pulsed current density of FSCC power management system reaches 14.3 A m$^{-2}$ as shown in Fig. 3e. Owning to the multifold increase in output charges and super-short discharging time (<1 ms), the current density is boosted by 807 times compared with 17.7 mA m$^{-2}$ of TENG (Supplementary Fig. 12b). The pulse power density reaches 954 W m$^{-2}$ on 2.4 kΩ load (Fig. 3f), which is elevated by 192 times as that of 4.97 W m$^{-2}$ of TENG on 20 MΩ (Supplementary Fig. 12c), almost twice larger than that of the previous highest record (500 W m$^{-2}$)[10]. The matching impedance is effectively reduced after the power management, which is more suitable for driving common electric devices (1–100 KΩ). With the increase of frequency, the pulse power density on 3 kΩ load increases from 403 W m$^{-2}$ at 0.1 Hz to 960 W m$^{-2}$ at 2 Hz, which is illustrated by Supplementary Fig. 12d. To reach the maximum pulse power output, some preliminary researches are carried out to discuss the influence of total voltage drop, types of the integrated substrate, capacitance of charge storage capacitor and different orders on current density to get optimum parameters, as shown in Supplementary Fig. 13 and Supplementary Note 4. And the voltage cumulative effect is proved again by comparing the voltage on different loads with the corresponding open-circuit output voltage of FSCC (Supplementary Fig. 14).

Total energy transfer efficiency of TENG power management is discussed in the following experiments. Higher energy transfer efficiency usually means larger output energy of power

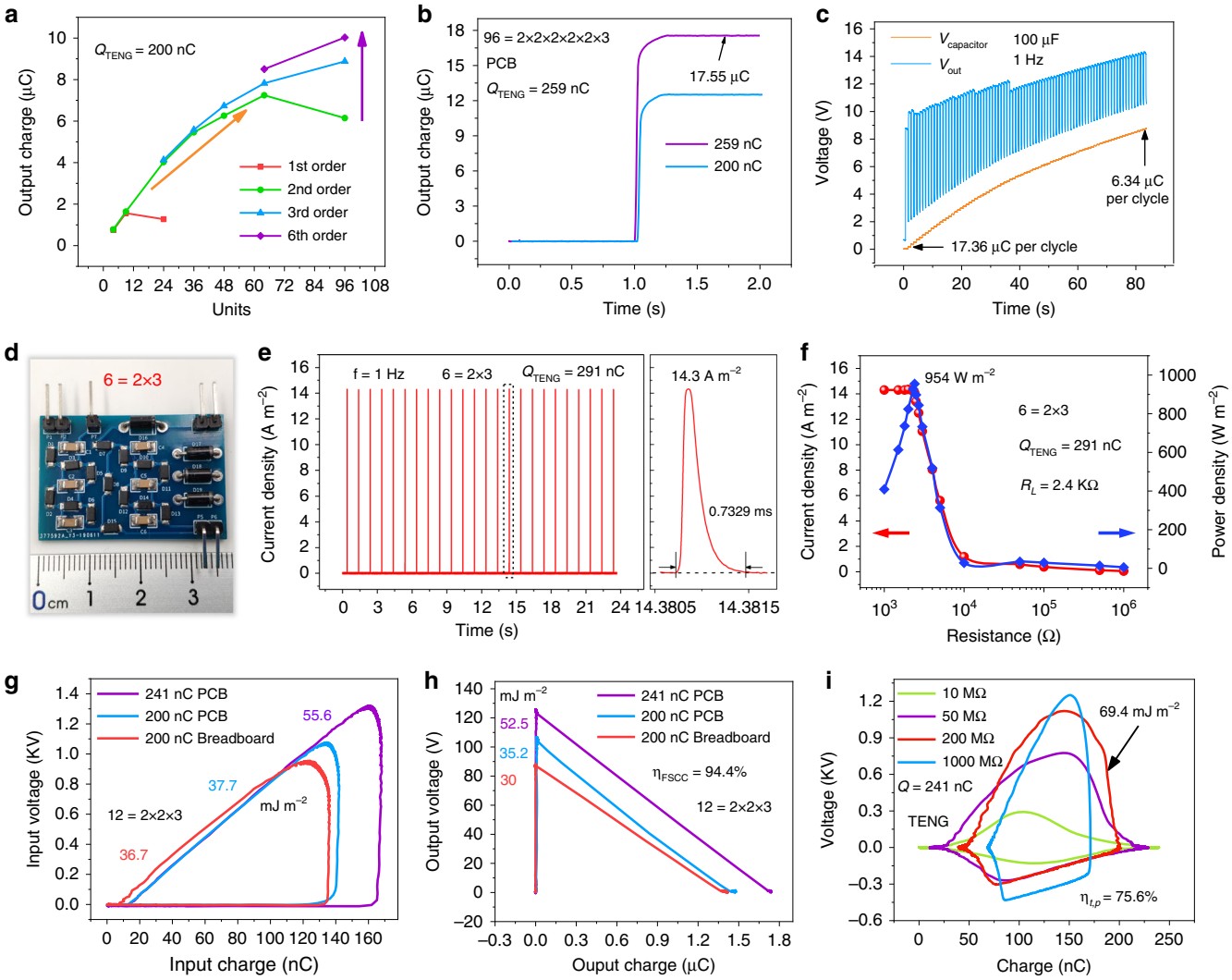

**Fig. 3 Maximizing pulse output with FSCC power management system. a** The output charge of FSCC with different orders and units per cycle (components are integrated on breadboard) at driving frequency of 1 Hz and TENG area of 20 cm⁻². **b** The maximum output charges of 96-unit FSCC per cycle, indicating 67.8 times charge boosting. **c** The voltage cumulative effect of FSCC that the open-circuit output voltage can increase with the increase in load voltage. **d** A photograph of 6 = 2 × 3 FSCC integrated on a printing circuit board (PCB). **e** The output current density of 6-unit FSCC, achieving a super-high current density of 14.3 A m⁻² per cycle. **f** The maximum power output of FSCC power management system per cycle, obtaining 192 times power boosting. **g** Energy input into FSCC per cycle from TENG under different charge quantities. **h** The maximum output energy of FSCC per cycle under different charge quantities in TENG. **i** Energy consumption per cycle on different resistances provided by TENG directly.

management system. Here, 1 nF charge storage capacitors in FSCC are employed to achieve large energy output (Supplementary Fig. 15a, b). First, we research the energy conversion efficiency of 12 = 2 × 2 × 3 FSCC itself, as shown in Fig. 3g, h. With 12 = 2 × 2 × 3 FSCC (Supplementary Fig. 15c–e), there is 55.56 mJ m⁻² energy density input into FSCC per cycle, and 52.5 mJ m⁻² energy density can output from it at 241 nC charge quantity, where the energy conversion efficiency of 12 = 2 × 2 × 3 FSCC can reach 94.3% calculated by Eq. (7), and it is used to describe the efficiency of FSCC itself only and belongs to efficiency of part. The detailed derivation process is shown in Supplementary Note 1.

$$\eta_{FSCC} = \frac{E_{out}}{E_{in}} = \frac{\int V_{out} dQ_{out}}{\oint V_{in} dQ_{in}} \quad (7)$$

$$\eta_{t,p} = \frac{E_{out,max}}{E_{R,opt}} = \frac{\int V_{out,max} dQ_{out,max}}{\oint V_{R,opt} dQ_{R,opt}} \quad (8)$$

As for the total efficiency of the power management system (total energy transfer efficiency) with the pulse output, which is used to describe the efficiency of the energy transfer process from TENG to load through FSCC power management system, we adopt method defined in work of Zi et al.[33] (Supplementary Note 5 and Fig. 3i), and total energy transfer efficiency reaches 75.6% calculated by Eq. (8).

**Constant output of FSCC power management system**. As we know, small electric devices used in our daily life are driven by a constant voltage power source, consequently, a constant output is necessary for power management system of TENG. We choose 96 = 2 × 2 × 2 × 2 × 2 × 3 FSCC for the constant output at 259 nC charge quantity with consideration of a large charge quantity needed for driving electric devices.

The constant output FSCC power management system and standard circuit for TENG are shown in Fig. 4a, b, respectively. A filter storage capacitor of 470 μF is connected in parallel at the output port of FSCC to keep a constant output, with the aim to fit

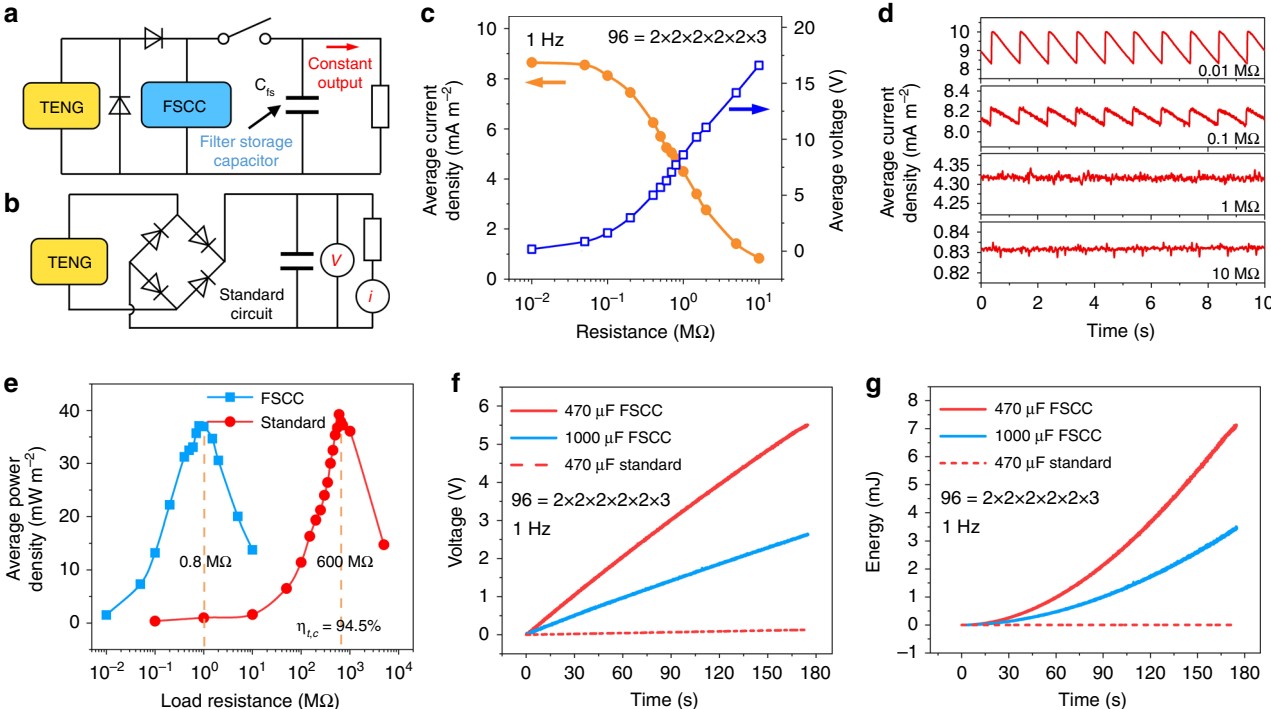

**Fig. 4 Maximizing constant output with FSCC power management system. a** The $96 = 2 \times 2 \times 2 \times 2 \times 2 \times 3$ FSCC power management system with constant output. The filter storage capacitor is 470 μF. **b** The standard circuit with the constant output for TENG. The filter storage capacitor is 1 μF. **c** The average current density and voltage at different loads with FSCC power management. **d** The current waveform at different loads with FSCC power management. **e** The average power density of FSCC power management and the standard circuit. The total energy transfer efficiency of 94.5% is achieved. **f** Charging capacitors and **g** Corresponding cumulative energy.

pulse and storage charge to maintain a stable voltage. Supplementary Fig. 16 shows the current-time curve on 0.01 MΩ load. Figure 4c gives the average current density and corresponding average voltage on different loads under the constant output. With increase in loads from 10 kΩ to 10 MΩ, average current density decreases from 8.65 to 0.83 mA m$^{-2}$ and voltage increases from 0.172 to 16.6 V. The unique voltage cumulative effect of FSCC benefits a slower current decay with a load increase, and a wider impedance applicability compared with other power management systems[23,29,34]. The detailed waveform of the average current on different loads is shown in Fig. 4d, from which the current tends to be constant with the voltage fluctuation of only 0.04 V when the load exceeds 0.1 MΩ. The energy transfer efficiency of constant output is also given by experiments (Fig. 4e). At last, a total energy transfer efficiency of 94.5% is achieved by comparing the maximum average power of FSCC power management system (37.09 mW m$^{-2}$ on 0.8 MΩ) with standard circuit (39.24 mW m$^{-2}$ on 600 MΩ).

$$\eta_{t,c} = \frac{P_{out,avg}}{P_{R,opt,avg}} = \frac{I_{out,avg} V_{out,avg}}{(V_{R,opt,avg})^2 / R} \qquad (9)$$

To compare with pervious works, the widely used method to calculate total energy transfer efficiency of constant output reported by Niu et al.[23] for TENG power management system is adopted here (Eq. (9) and Supplementary Note 6). For a further comparison of the standard circuit with FSCC power management, by charging for 175 s, voltage of 470 μF capacitor increases to 0.123 V with energy of 3.6 μJ for the standard circuit. Instead, a 470 and 1000 μF capacitor can be charged to 5.51 and 2.63 V with the stored energy of 7.141 and 3.494 mJ with FSCC (Fig. 4f, g),

respectively. Obviously, the improvement of capacitor voltage by 44.8 times is reached.

**Applications of FSCC power management system for TENG**. In order to demonstrate the performance of FSCC power management system, some applications are carried out in following experiments. In all, 160 green LEDs (5 mm) in parallel and 2 green LEDs (10 mm) in serial are lighted brightly by the system with 96-unit FSCC both in dark and light environment (Fig. 5a, b, Supplementary Fig. 17a and Movie 2). In comparison, LEDs above are powered by TENG with half-wave circuit and only two LEDs (10 mm) can be faintly lighted. Here, we also light LEDs above by $6 = 2 \times 3$ FSCC (Supplementary Fig. 17b). However, the super-short luminescence time leads to a weak visual brightness, in other words, the exposure time is too short to obtain the real brightness. Figure 3e and Supplementary Fig. 17c display the discharging time of 96- and 6-unit FSCC. Buzzer is a kind of sound electric device and the sound of people communication commonly is about 50–60 dB. Over 70 dB sound is generated from the buzzer driven by the system with 96-unit FSCC, as shown in Fig. 5c and Supplementary Movie 3. The above results fully demonstrate the large current and power performance of FSCC power management. As for the constant output, a medium size digital vernier caliper is powered by the system to measure length parameter continuously, at operating voltage of 2.57 V and frequency of 1 Hz (Fig. 5d–f and Supplementary Movie 4). As shown in Fig. 5g–i and Supplementary Movie 5, a temperature hygrometer with three sensors at working voltage over 1.5 V driven by TENG at operating frequency of 1.5 Hz with the FSCC power management system, to measure the temperature and humidity information of the environment around continuously. Supplementary Fig. 18 shows the average driving current of the digital vernier caliper (14 μA at 2.6 V) and temperature

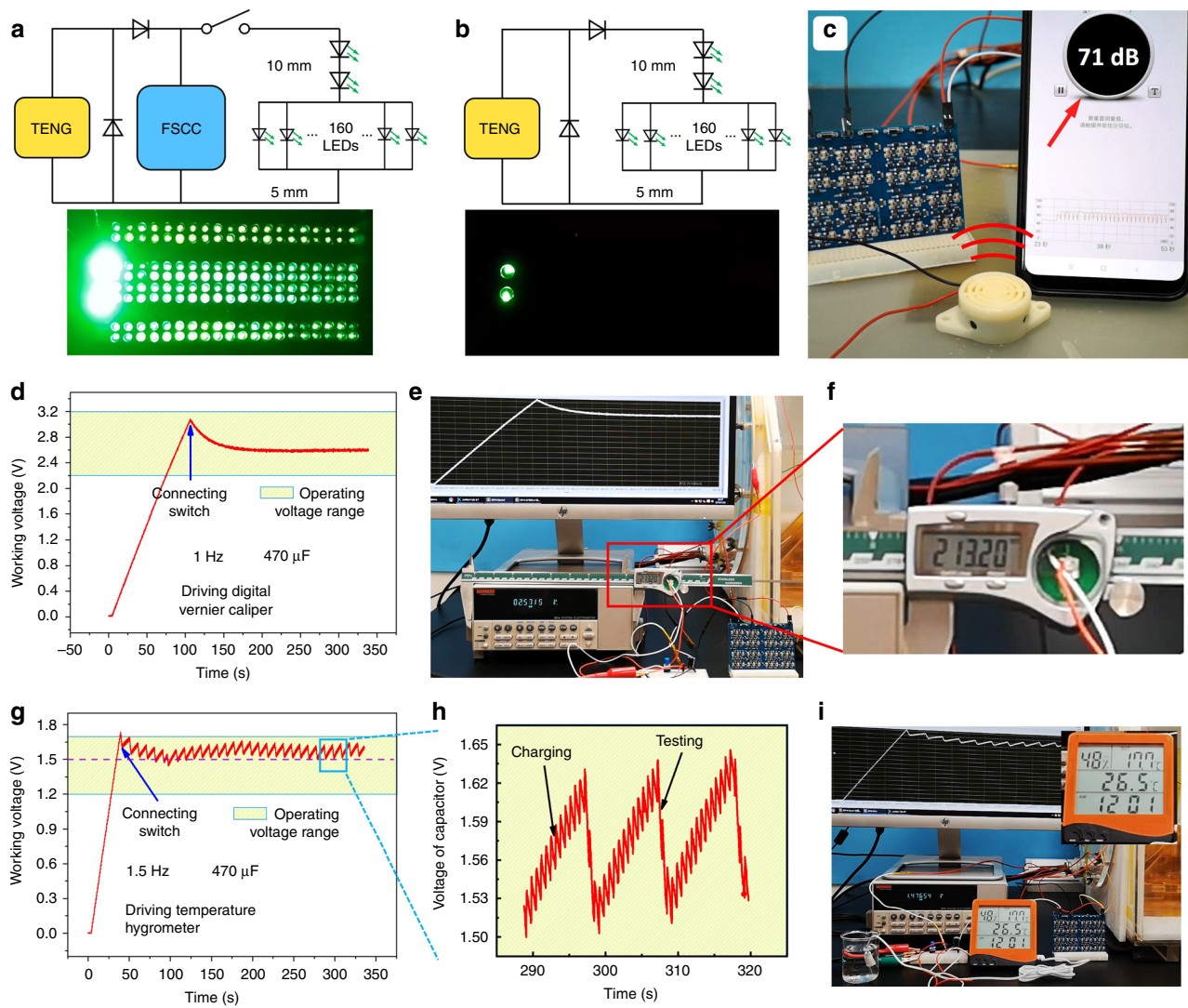

**Fig. 5 Applications of FSCC power management for TENG.** Lighting 160 green LEDs with 5 mm diameter in parallel and 2 green LEDs with 5 mm diameter in serial by FSCC power management system (**a**) and TENG (**b**) directly at 1 Hz, the circuit diagram used to light LEDs is above the optical image**. c** Driving a buzzer with the pulse output mode at 1 Hz. **d** Voltage-time curve for continuously driving a medium size digital vernier caliper and **e** the corresponding image. **f** the enlarged image of the selected area in **e**. **g** Voltage-time curve of a temperature hygrometer with three sensors driven by 96-unit FSCC power management system. **h** The detailed voltage curve of the selected area in **g**. **i** The actual image of driving the temperature hygrometer.

hygrometer (24.6 µA at 1.6 V) by a constant voltage source. In addition, the stability performance of 96-unit FSCC is also tested (Supplementary Fig. 19), and after more than 30,000 cycles at 1 Hz, the output voltage finally maintains at 8.8 V from the initial 9.4 V. It is believed that FSCC with such a quite stable output can meet the requirements of practical applications.

Applications above strongly prove the high conversion performance of FSCC power management, and the great potential for the double-function output power management in satisfying various demands and internet of things. It also indicates a significant guidance on the next stage development toward multi-functional output power management system.

## Discussion

In summary, the creative FSCC is designed by fractal configuration evolution to reduce the negative impact of total output voltage drop across the switches and effectively convert high voltage and low charges into low voltage and high charges. Compared with a traditional transformer, the FSCC has the advantages of no electromagnetic noise, minimum output

impendence and high electrostatic voltage applicability. Moreover, a high efficiency FSCC power management system with double-function output modes is also established for TENG, by which, over 67 times charge boosting and 954 W m$^{-2}$ power density are achieved under the pulse output, and over 94% energy transfer efficiency is realized under a constant output, setting the records for TENG. The concept of multi-function output mode after power management is proposed for TENG to satisfy demands in various cases. It is of a significant guidance for the development of power management of TENG toward multi-function output in the next stage. The successful demonstrations of FSCC applied in TENG also manifest the great potential for FSCC to be applied in the electric vehicle, DC micro-grid area etc.

## Methods

**Fabrication of the TENG**. Two pieces of acrylic board were cut into $3 \times 45 \times 68$ mm$^3$ substrates by laser cutter (JKJG-MEIBA-L-4060). A $3 \times 45 \times 68$ mm$^3$ foam with $5 \times 5$ mm$^2$ gap was adhered to the top surface of bottom acrylic substrate, a 33 mm × 61 mm × 25 µm Al electrode (Bottom) with 10 mm chamfer was adhered to a 1 mm acrylic board which is adhered to the foam, a 50 µm × 45 × 68 mm PTFE film was pasted on the Al electrode by 100 µm tape. The upper Al electrode that the

same as the bottom one was adhered to up acrylic substrate. Finally, the contact switch was mounted on the top and bottom substrates corresponding to the gap.

**Fabrication of the FSCC.** Integrating on breadboard: the diode model used in different FSCCs is 1N4007. The FSCCs were installed on breadboards with size of $35 \times 47$ mm$^2$, $53 \times 82$ mm$^2$, and $55 \times 85$ mm$^2$ based on the order and units. Integrating on PCB: the surface mounted devices diode model is 1N4007W, the package specifications for SMD diode and capacitor is 1206, the breakdown voltage for 1 nF and 22 nF capacitor are 2 kV and 200 V, respectively. The PCB schematic diagram was designed by Altium Designer, and PCB proofing patches were completed by commercial factories.

**Measurement.** The contact-separation process of TENG was driven by a linear motor (LINMOT E1200-P01) under sinusoidal motion in an acrylic glove box with 5% relative humidity. The voltage was measured by an electrostatic voltmeter (TREK 370-3, voltage over 200 V) and electrometer (Keithley 6514, voltage below 200 V), the current and charge were measured by electrometer (Keithley 6514). The voltage driving electric devices was measured by the electrometer as well. A programmable power supply (Keithley 2230G) was used to drive the electric devices to test the driving current and turn-on voltage drop with electrometer. Q–V curve (Fig. 3g, i) was measured by Keithley 6514 (charge) and TREK 370-3 (voltage) at the same time through dual channel testing system, with the same sampling rate of 1000 data per second and no filtering or smoothing processing. Q–V curve (Fig. 3h) was measured by two 6514 at the same time through dual channel testing system, which measured Q and V, respectively. The grounding terminal of the measuring line of the two testing devices were connected together to gain accurate data.

## Data availability
The authors declare that the main data supporting the findings of this study are available within the article and its Supporting Information files. Extra data are available from the corresponding authors on reasonable request.

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

## Acknowledgements

This work was supported by National Key R & D Project from Minister of Science and Technology (2016YFA0202704), National Natural Science Foundation of China (NSFC) (51572040), the Chongqing University Postgraduates' Innovation Project (grant no. CYB18061), and the Fundamental Research Funds for the Central Universities (2019CDXZWL001).

## Author contributions

C.H. and Z.L.W. supervised the project. W.L., Z.W., C.H., H.G., and Z.L.W. conceived the project and designed the experiment part. W.L. and Z.W. fabricated the devices and completed the electrical performance measurement part. G.W. took charge the design and fabrication of PCB part. Q.Z., W.H., L.L., X.W., and Y.X., helped plot and analyze data. W.L., C.H., and H.G. wrote the paper. All authors contributed to the paper.

## Competing interests

The authors declare no competing interests.
