## [Peer Review File · Nature Communications]

Reviewers' comments:

Reviewer #1 (Remarks to the Author):

This manuscript showed the fractal design based switched-capacitor-converters as power management circuit for triboelectric nanogenerator. With using this circuit, the pulsed power density was elevated by 192 times than without this circuit and 94.5% of an energy transfer efficiency was achieved. However, some modification needs to be conducted for accepting to this Nature Communications.

1. There is no analysis or discussion of Fig 3e, f in the results part.
2. The explanation of how to draw Q-V curve in Fig 3g-i would be added.
3. Supplementary Figure 12, there is no reason why 1 nF capacitor showed highest output. In Supplementary Figure 13, here also no discussion of this figure.
4. On Fig 5b, some comments are needed to be added for easy understanding of what correspond to each state of with/out using the circuit.
5. There are many punctuation errors and some commas should be change to period.

Reviewer #2 (Remarks to the Author):

Nowadays, large energy output and energy transfer efficiency are two key factors for TENG toward actual applications. In this manuscript, authors put forward a fractal design based switched-capacitor-convertors (FSCC), by which, the highest record of peak power density and over 94% total energy transfer efficiency of TENG are achieved. This is an interesting work with enough innovation, and the manuscript is well-organized with a lot of supporting information for readers. Therefore, it is good enough to be published in Nature Communication. However, there are a few minor questions that need to be addressed before this manuscript accepted.

1. A power management system with stable output is necessary for the actual applications of TENG, so the time curve is suggested to demonstrate the reliability issue.
2. The concepts of total energy transfer efficiency and energy conversion efficiency in manuscripts are confusing, authors should clarify these concepts for readers.
3. The fractal design based switched-capacitor-convertors is interesting, is there a limitation of the orders for this design? What would happen if the orders is too large?
4. The combination of this FSCC design and contact-separation mode TENG shows a quite good result, is it a must to combine FSCC design and contact-separation mode TENG? Can other mode of TENG have a good power management characteristic with this FSCC?
5. Charge storage capacitors with the range of 1 nF to 22 nF are used in SCC and FSCC, to have a better understand of this work for readers, authors should give an explicit explain toward how to choose these charge storage capacitors.

Reviewer #3 (Remarks to the Author):

This work reports switched-capacitor-convertors for dual-function output power management of triboelectric nanogenerator (TENG), where the fractal designed configurations were proposed. This power management circuit show attractive features of high conversion efficiency, minimum output impedance, electrostatic voltage applicability, and high step-down ratio. After the comprehensive study, the developed circuit achieved higher than 94% total energy transfer efficiency. The research topic of this work is very interesting, the proposed method is innovative, and the

manuscript is well organized. Thus, it is recommended to the acceptance.

1. Fig. 1 could be improved better. It is suggested to remove Fig. 1(b) to the Supplementary file.
2. It is suggested to replace Fig. 2(g) with the photograph of the fabricated TENG.
3. Why the current curve in Fig. 3(e) is dissymmetric?
4. The photographs in Fig. 5(d,e) should be enlarged to show more details.
5. More related articles of TENG should be cited to help readers easily understand the contribution of this work.

Point-to-Point Response to the Reviewer's Comments

(Comments in black, response in blue , corrections in yellow background)

Dear reviewers:

Thank you for your detailed and useful comments and suggestions on our manuscript. We have revised the manuscript accordingly and the detailed corrections are listed below point by point.

Reviewer #1 (Remarks to the Author):

This manuscript showed the fractal design based switched-capacitor-converters as power management circuit for triboelectric nanogenerator. With using this circuit, the pulsed power density was elevated by 192 times than without this circuit and 94.5% of an energy transfer efficiency was achieved. However, some modification needs to be conducted for accepting to this Nature Communications.

Response: We highly appreciate the reviewer's positive comments and good suggestions on our work. And we also thank the reviewer's detailed and responsible reviewing of our work.

1. There is no analysis or discussion of Fig 3e, f in the results part.

Response: Thanks for your valuable suggestion. Figure 3e and 3f show the maximum pulsed current density and pulsed power density curve with optimum FSCC (6=2×3) power management system, and 807 times and 192 times enhancement of current density and pulsed power density are reached respectively compared with that of TENG without any power management. Here, owing to the multifold increase in output charges achieved by FSCC and super-short discharging time (< 1ms) controlled by contact switch, 14.3 A m⁻² pulsed current density and 954 W m⁻² pulsed power density are realized for the first time. The result is almost twice larger than that of the previously highest record (500 W m⁻², Adv. Mater. 2014, 26, 3788–3796). Besides, the matching impedance decreases from 20 MΩ to 2.4 KΩ with FSCC, which is more suitable for driving common electric devices (Inner impedance:1~100 KΩ). To obtain the maximum current density, some works for optimizing parameters are also carried out, including the investigation in influence of total voltage drop, charge quantity of TENG, types of the integrated substrate, capacitance of charge storage capacitor and different orders on current density, as shown in Supplementary Figure 13 and Note 4. It is clear that FSCC makes TENG have the characteristics of high current density, high power density, small matching impedance and large charge output, leading TENG closer to practical application. The revised discussions of Fig. 3e and 3f in manuscript are listed below:

Besides, when the charge quantity of TENG is 291 nC at frequency of 1 Hz and the energy storage capacitor is 1 nF, the short-circuit pulsed current density of FSCC power management system reaches 14.3 A m⁻² as shown in **Fig. 3e**. Owing to the multifold increase in output charges and super-short discharging time (< 1ms), the current density is boosted by 807 times compared with 17.7 mA m⁻² of TENG (**Supplementary Figure 12b**). The pulse power density reaches 954 W m⁻² for the first time on 2.4 kΩ load (**Fig. 3f**), which is elevated by 192 times as that of 4.97 W m⁻² of TENG on 20 MΩ (**Supplementary Figure 12c**), almost twice larger than that of the previous highest record (500 W m⁻²)⁹. The matching impedance is effectively reduced after the power management, which is more suitable for driving common electric devices (1~100 KΩ). With the increase of frequency, the pulse power density on 3 kΩ load increases from 403 W m⁻² at 0.1 Hz to 960 W m⁻² at 2 Hz, which is illustrated by **Supplementary Figure 12d**. To reach the maximum pulse power output, some preliminary researches are carried out to discuss the influence of total voltage drop, types of the integrated substrate, capacitance of charge storage capacitor and different orders on current density to get optimum parameter, as shown in **Supplementary Figure 13 and Note 4**.

2. The explanation of how to draw Q-V curve in Fig 3g-i would be added.

Response: Thanks for your good suggestion. The Q-V curve first proposed by Zi at al. is used to evaluate the output energy of TENG in one operating cycle (Nat. Common. 2015, 6, 8376).

$$E = \oint V dQ$$

For the Q-V curve in this work, two testing equipment are used to measure voltage and charge at the same time. Here, an electrostatic voltmeter (TREK 370-3) which has ±3 kV measuring range is used to test the voltage over 200V (Fig. 3g and 3i), and a Keithley 6514 is used to measure the voltage below 200 V (Fig. 3h). Another Keithley 6514 is used to measure the charge quantity. In brief, Q-V curve was acquired by 6514 to measure charge and TREK 370-3 (or 6514) to measure voltage at the same time through dual channel testing system with the same sampling rate of 1000 data per second. The grounding terminal of the measuring line of the two testing equipment should be connected together, to gain the accurate data. Especially, smoothing or filtering the test data may cause serious distortion of the waveform, therefore, filtering or smoothing should not be used when testing Q-V curve. We can draw the Q-V curve in Origin software by setting Q as X-axis and V as Y-axis, and the energy value is the area surrounded by Q-V curve. In addition, we have added this information in the method part.

The contact-separation process of TENG was driven by a linear motor (LINMOT E1200-P01) under sinusoidal motion in an acrylic glove box with 5% relative humidity. The voltage was measured by an electrostatic voltmeter (TREK 370-3, voltage over 200 V) and electrometer (Keithley 6514, voltage below 200 V), the current and charge were measured by electrometer (Keithley 6514), the voltage

driving electric devices was measured by electrometer as well. A programmable power supply (Keithley 2230G) was used to drive the electric devices to test the driving current and turn-on voltage drop with electrometer. Q-V curve (Fig. 3g and 3i) was measured by Keithley 6514 (charge) and TREK 370-3 (voltage) at the same time through dual channel testing system, with the same sampling rate of 1000 data per second and no filtering or smoothing processing. Q-V curve (Fig. 3h) was measured by two 6514 at the same time through dual channel testing system which measured Q and V, respectively. The grounding terminal of the measuring line of the two testing devices were connected together to gain accurate data.

3. Supplementary Figure 12, there is no reason why 1 nF capacitor showed highest output. In Supplementary Figure 13, here also no discussion of this figure.

Response: Thank the reviewer for the detailed and valuable comments. The discharging process of FSCC is equivalent to the discharging process of a big capacitor, which can be described by equation as follow:

$$i = \frac{Q_{in}}{RC_s} e^{-\frac{t}{RNC_s}}$$

In the equation above, $e^{-\frac{t}{RNC_s}} \in (1 \sim 0)$ is a decay factor. So we can get the maximum value of current: $i_{max} = \frac{Q_{in}}{RC_s}$, and we can see that the maximum current is

inversely proportional to the capacitance of charge storage capacitor, so the smaller charge storage capacitor leads to higher output current. Here, experimental result in Supplementary Figure 13b (reordered from Supplementary Figure 12b) is also consistent to the theoretical analysis, current with 1nF capacitor is higher than that with capacitor over 1 nF. I am sorry that we have no capacitor smaller than 1nF, so the current value is cut off at 1nF. The revised explanation has been added in Supplementary Note 4 (Response to “Supplementary Figure 12, there is no reason why 1 nF capacitor showed highest output”). In Supplementary Figure 14 (reordered from Supplementary Figure 13), the voltage on external load and open-circuit output voltage of FSCC increase along with the increasing of load, naming the voltage cumulative effect, which is caused by the accumulated residual charge of charge storage capacitors ($V=Q/C$) in each discharge cycle, and the residual charge is owing to that the external load is too large to release all the charge. The open-circuit voltage of FSCC increases from 9.9 V to 14.3 V when FSCC charging a capacitor as shown in Fig. 3c, which also demonstrates the voltage cumulative effect. The discussion of voltage cumulative effect described in Fig. 3c, and Supplementary Figure 14, is to prove this effect again from driving external load aspect. (Response to “In Supplementary Figure 13, here also no discussion of this figure.”)

Supplementary Note 4 | Preliminary research for maximum pulse power output.

The discharging process of FSCC is equivalent to discharging process of a big capacitor, which can be described by equation below:

$$i = \frac{U_0}{R} e^{-\frac{t}{RC}} \quad (12)$$

Where C and U_0 is respectively the capacitance and the voltage of capacitor. So the discharging process of FSCC can be conducted according to Equation 4 ($Q_{out} = NQ_{in}$) and 6 ($Q_{out} = V_{out} \cdot C_s N$).

$$i = \frac{Q_{in}}{RC_s} e^{-\frac{t}{RNC_s}} \quad (13)$$

In the equation above, as $e^{-\frac{t}{RNC_s}} \in (1 \sim 0)$ is a decay factor the maximum value of current is:

$$i_{max} = \frac{Q_{in}}{RC_s} \quad (14)$$

Therefore, the maximum current is inversely proportional to the capacitance of charge storage capacitor.

Preliminary researches are carried out with TENG of 200 nC charge quantity for quantitative characterization. Firstly, the influence of voltage drop is discussed in **Supplementary Figure 13a**, and total voltage drop could reduce output current linearly. **Supplementary Figure 13b** shows that small (1 nF) charge storage capacitor is suitable to obtain high current in experimental aspect, which is consistent to theoretical analysis for the influence of voltage drop. The current output under different units indicates that TENG with 5-unit FSCC has a maximum current output (**Supplementary Figure 13c**). **Supplementary Figure 13d** deeply discusses the impacts of orders on current, and shows that 6=2×3 and 8=2×2×2 FSCC can realize maximum output current. At last, we chose 6=2×3 FSCC for considering its smaller size in experiment. Finally, when FSCC is integrated on breadboard and on PCB (**Supplementary Figure 13e-f**), the maximum pulse power reaches 247 W m⁻² and 340 W m⁻² at 200 nC, respectively.

In addition, when charging a 100 μF capacitor with FSCC on PCB, the voltage of capacitor can reach 8.76 V from 0 V within 85 seconds. **Meanwhile, the output voltage of FSCC increases from 9.9 V to 14.3 V as shown in Fig. 3c**, which indicates a unique voltage cumulative effect and better output stability compared with works based on inductance transducer. And the voltage cumulative effect is caused by the accumulated residual charge of charge storage capacitors ($V=Q/C$) in each discharge cycle, and the residual charge is owing to that the external load is too large to release all the charge.

4. On Fig 5b, some comments are needed to be added for easy understanding of what correspond to each state of with/without using the circuit.

Response: Thanks for the good suggestion. For easy understanding of Fig 5b, we have reorganized this figure, and separate the photography of lighting LEDs with/without the circuit, with the corresponding circuit above them. The revised figure is as follows,

Fig. 5 | Applications of fractal design based switched-capacitor-converter power management for triboelectric nanogenerator. Lighting 160 green LEDs with 5 mm diameter in parallel and 2 green LEDs with 5 mm diameter in serial with FSCC power management system a and without FSCC b at 1 Hz, the circuit diagram used to light LEDs is above the optical image. c, Driving a buzzer with the pulse output mode at 1 Hz.

5. There are many punctuation errors and some commas should be changed to periods.

Response: Thanks the reviewer for the good suggestion. In the revised manuscript, we have carefully checked the contents, and there are 16 places where commas were changed to periods. After the related punctuation errors are revised, it is easier to read and understand this paper.

Reviewer #2 (Remarks to the Author):

Nowadays, large energy output and energy transfer efficiency are two key factors for TENG toward actual applications. In this manuscript, authors put forward a fractal design based switched-capacitor-convertors (FSCC), by which, the highest record of peak power density and over 94% total energy transfer efficiency of TENG are achieved. This is an interesting work with enough innovation, and the manuscript is well-organized with a lot of supporting information for readers. Therefore, it is good enough to be published in Nature Communication. However, there are a few minor questions that need to be addressed before this manuscript accepted.

Response: We highly appreciate the reviewer's positive and valuable comments on our work as "enough innovation". And we also thank the reviewer's detailed and responsible reviewing of our work.

1. A power management system with stable output is necessary for the actual applications of TENG, so the time curve is suggested to demonstrate the reliability issue.

Response: Thanks for your instructive suggestion. Stable output is important in actual

applications, and we have tested the stability performance of the power management circuit in the revised manuscript. Because output charge is accumulated in one direction, so, the charge quantity would beyond the measuring range. Therefore, output voltage is chosen to characterize the stability performance of FSCC power management system at 1 Hz. Power management circuit of $96=2\times2\times2\times2\times2\times3$ FSCC integrated on PCB is used, the corresponding data figure is shown below. After more than 30000 cycles, the output voltage finally maintains at 8.8 V from the initial 9.4 V, which is a quite good result and can meet the requirements of actual applications.

Supplementary Figure 19 | The stability performance of $96=2\times2\times2\times2\times2\times3$ FSCC.

In addition, the stability performance of 96-unit FSCC is also tested (**Supplementary Figure 19**), and after more than 30000 cycles at 1 Hz, the output voltage finally maintains at 8.8 V from the initial 9.4 V. It is believed that FSCC with such a quite stable output can meet the requirements of practical applications.

2. The concepts of total energy transfer efficiency and energy conversion efficiency in manuscripts are confusing, authors should clarify these concepts for readers.

Response: Thanks for your good suggestion. The total energy transfer efficiency is used to describe the efficiency of the energy transfer process from TENG to load through FSCC power management system, which belongs to efficiency of whole energy system. As for the energy conversion efficiency, it is used to describe the efficiency of FSCC itself only and belongs to efficiency of part.

3. The fractal design based switched-capacitor-convertors is interesting, is there a limitation of the orders for this design? What would happen if the orders is too large?

Response: Thanks for the valuable question. The detailed structure of FSCC is described with expanded form below:

$$N_L = m_L \times m_{L-1} \cdots \cdots m_2 \times m_1 \quad (N \geq 2)$$

Where N_L is the total basic unit number of FSCC, and m_i is the unit number of per order. When the basic unit number N_L is given, it is obvious that the expansion of N_L has a finite number of terms, meaning that there is a limitation of orders with

given N_L . (Response to “is there a limitation of the orders for this design”). There are two cases existing when the order is too large. One is that the output charge, current and energy conversion efficiency all increase toward the beneficial side with the increase in order when basic unit number N_L is given. The other is that the total basic unit number N_L will increase and the output voltage will decrease along with the order increase when N_L is not limited. And the output voltage would be too low to drive electric devices if order number is too large. (Response to “What would happen if the orders is too large?”). Therefore, the basic unit number N_L is decided by the desired output voltage and voltage of TENG, and the large order is expected when N_L is determined.

4. The combination of this FSCC design and contact-separation mode TENG shows a quite good result, is it a must to combine FSCC design and contact-separation mode TENG? Can other modes of TENG have a good power management characteristic with this FSCC?

Response: Thanks for your valuable question. Previous work has proved that the TENG has extremely low energy utilization efficiency when directly driving electric devices (*Nat. Commun.* 2015, 6, 8975). The basic working principle of FSCC is charging the charge storage capacitors of FSCC in series state and the capacitors discharge in parallel, so that the FSCC can decrease the output voltage and increase the output charge of TENG. Besides, fractal design is used to effectively decrease the total voltage drop to achieve maximum output. Here, the TENG is equal to a voltage/charge source, and FSCC is equal to a transformer to output energy matching common device, thus, it is a must to combine FSCC design and TENG for achieving effectively energy usage in common electric devices (Response to “is it a must to combine FSCC design and contact-separation mode TENG”). The output performance after power management is fully depended on the power management circuit when the charge and voltage of TENG keep constant. Therefore, other modes TENG also can have a good power management characteristic with suitable structure FSCC power management system (Response to “Can other modes of TENG have a good power management characteristic with this FSCC?”).

5. Charge storage capacitors with the range of 1 nF to 22 nF are used in SCC and FSCC, to have a better understand of this work for readers, authors should give an explicit explain toward how to choose these charge storage capacitors.

Response: Thank the reviewer for this valuable question. From the result of Fig. 3 and Supplementary Note 4, we can know that small capacitance of charge storage capacitor is suitable when a high current and power is needed. Large charge quantity is needed when driving electric devices in constant current mode, thus, large capacitance of charge storage capacitor is suitable to obtain large charge quantity to drive electric devices when the output voltage meets the required value in constant mode.

Reviewer #3 (Remarks to the Author):

This work reports switched-capacitor-convertors for dual-function output power management of triboelectric nanogenerator (TENG), where the fractal designed configurations were proposed. This power management circuit show attractive features of high conversion efficiency, minimum output impedance, electrostatic voltage applicability, and high step-down ratio. After the comprehensive study, the developed circuit achieved higher than 94% total energy transfer efficiency. The research topic of this work is very interesting, the proposed method is innovative, and the manuscript is well organized. Thus, it is recommended to the acceptance.

Answer: We highly appreciate the reviewer’s positive comments on our work as “innovative”. And we also thank the reviewer’s detailed and responsible reviewing of our work.

1. Fig. 1 could be improved better. It is suggested to remove Fig. 1(b) to the Supplementary file.

Response: Thanks reviewer for the valuable suggestion. According to this suggestion, we have removed Fig. 1b to the Supplementary file as Supplementary Figure 1. We also checked the content and revised the related part to match the revised Figure. The revised Figure 1 is as follow:

Fig. 1 | A comparison of a conventional transformer with a fractal design based switched-capacitor-convertor. a, The operating mechanism of a conventional electromagnetic transformer, a basic switched capacitor convertor (SCC) and a fractal switched-capacitor-convertors (FSCC). **b,** The concept of fractal structure evolution in order increasing (upper) and similar design based fractal structure evolution for FSCC in order increasing (lower), which can be achieved by replacing capacitors in SCC

repeatedly with 2=2 SCC unit.

2. It is suggested to replace Fig. 2(g) with the photograph of the fabricated TENG. Response: Thanks for your helpful suggestion. We replaced Fig. 2g with the photograph of the fabricated TENG in the revised manuscript for intuitive understanding of TENG itself, and the original Fig. 2g with each layer structure of TENG has been removed to the Supplementary file as Supplementary Figure 5b. The revised Fig. 2g is as follows.

Fig. 2 | The principle of fractal design based switched-capacitor-converter power management system for triboelectric nanogenerator. a, Initialization state of 4=2x2 FSCC system. **b-c,** The step-down process during a charging/discharging cycle. The FSCC can achieve multifold charge output through the auto-switch of capacitors from serial to parallel connection. **d,** The equivalent diagram of discharging process of FSCC in **c.** The turn-on voltage drop of diode has a significant influence on charge output. **e,** The output voltage and **f,** the output charge of 24-unit FSCC with different orders. 1st order FSCC has the largest turn-on voltage drop and the output charge increases rapidly with multi-order FSCC. **g,** Photograph of triboelectric nanogenerator (TENG) with a contact switch in this work. **h,** Schematic diagram of the power management system, which consists of a TENG, half-rectifier circuit, contact switch and FSCC, and there are two output modes in pulse and constant after power management.

3. Why the current curve in Fig. 3(e) is dissymmetric?

Response: Thanks for your detailed comment. The output current of the FSCC power management system is rectified by a half-wave rectifier, therefore, the current curve is

asymmetric in up and down direction. As for the -asymmetric output current in left and right direction, the discharging process of FSCC is equivalent to discharging process of a big capacitor, which can be described by equation below:

$$i = \frac{U_0}{R} e^{-\frac{t}{RC}}$$

The current will decrease from maximum to zero during the discharge process, therefore, the current curve in Fig. 3e is asymmetric in left and right direction.

4. The photographs in Fig. 5(d, e) should be enlarged to show more details.

Response: Thanks for your good suggestion. In order to show more details, we have reorganized Figure 5d, e, and the photographs have been enlarged. The revised Figure is as follows.

Figure 5, d, Voltage-time curve for continuously driving a medium size digital vernier caliper and e, the corresponding image. f, the enlarged image of the selected area in e. g, Voltage-time curve of a temperature hygrometer with 3 sensors driven by 96-unit FSCC power management system. h, The detailed voltage curve of the selected area in e. i, The actual image of driving the temperature hygrometer.

5. More related articles of TENG should be cited to help readers easily understand the contribution of this work.

Response: Thanks for the valuable suggestion. We have cited more related articles of TENG in the manuscript. And the added references are listed as follow:

3. Cheng L, Xu Q, Zheng Y, Jia X, Qin Y. A self-improving triboelectric nanogenerator with improved charge density and increased charge accumulation speed. *Nat. Commun.* **9**, 3773 (2018).
20. Khandelwal G, Chandrasekhar A, Raj N, Kim SJ. Metal-Organic Framework: A Novel Material for Triboelectric Nanogenerator-Based Self-Powered Sensors and Systems. *Adv. Energy Mater.* **9**, 8 (2019).

21. Lai Y-C, *et al.* Actively Perceiving and Responsive Soft Robots Enabled by Self-Powered, Highly Extensible, and Highly Sensitive Triboelectric Proximity- and Pressure-Sensing Skins. *Adv. Energy Mater.* **30**, 1801114 (2018).
22. Luo J, *et al.* Flexible and durable wood-based triboelectric nanogenerators for self-powered sensing in athletic big data analytics. *Nat. Commun.* **10**, 5147 (2019).
23. Parida K, Xiong JQ, Zhou XR, Lee PS. Progress on triboelectric nanogenerator with stretchability, self-healability and bio-compatibility. *Nano Energy* **59**, 237-257 (2019).
24. Zhang C, *et al.* Conjunction of triboelectric nanogenerator with induction coils as wireless power sources and self-powered wireless sensors. *Nat. commun.* **11**, 58-58 (2020).
41. Cao Z, Wang S, Bi M, Wu Z, Ye X. Largely enhancing the output power and charging efficiency of electret generators using position-based auto-switch and passive power management module. *Nano Energy* **66**, 104202 (2019).
42. Hang X, *et al.* Triboelectric Nanogenerator Networks Integrated with Power Management Module for Water Wave Energy Harvesting. *Adv. Funct. Mater.* **29**, 1807241 (2019).
43. Xia X, Wang H, Basset P, Zhu Y, Zi Y. Inductor-Free Output Multiplier for Power Promotion and Management of Triboelectric Nanogenerators toward Self-Powered Systems. *ACS appl. Mater. inter.* **12**, 5892-5900 (2020).
44. Xu S, Zhang L, Ding W, Guo H, Wang X, Wang ZL. Self-doubled-rectification of triboelectric nanogenerator. *Nano Energy* **66**, 104165 (2019).

REVIEWERS' COMMENTS:

Reviewer #1 (Remarks to the Author):

The authors have shown the detailed explanations and modification through the last review. In results and discussion and experimental parts, thank you for offering discussions for experimental results and the method to extract the graph. Some errors are well corrected and it is now acceptable to publish in Nature Communications.

Reviewer #2 (Remarks to the Author):

The questions are answered in detail.

Reviewer #3 (Remarks to the Author):

Since the authors have addressed most of reviewers' comments and the manuscript has been greatly strengthened, it is recommended to the acceptance.

Point-to-Point Response to the Reviewer's Comments

(Comments in black, response in blue)

Reviewer #1 (Remarks to the Author):

The authors have shown the detailed explanations and modification through the last review.

In results and discussion and experimental parts, thank you for offering discussions for experimental results and the method to extract the graph. Some errors are well corrected and it is now acceptable to publish in Nature Communications.

Response: Thanks for your strong efforts and valuable comments on our work.

Reviewer #2 (Remarks to the Author):

The questions are answered in detail.

Response: Thanks for your strong efforts and valuable comments on our work.

Reviewer #3 (Remarks to the Author):

Since the authors have addressed most of reviewers' comments and the manuscript has been greatly strengthened, it is recommended to the acceptance.

Response: Thanks for your strong efforts and valuable comments on our work.